# Primer synthesis by a eukaryotic-like archaeal primase is independent of its Fe-S cluster

Sandro Holzer[1], Jiangyu Yan[2], Mairi L. Kilkenny[1], Stephen D. Bell [2,3] & Luca Pellegrini [1]

DNA replication depends on primase, the specialised polymerase responsible for synthesis of the RNA primers that are elongated by the replicative DNA polymerases. In eukaryotic and archaeal replication, primase is a heterodimer of two subunits, PriS and PriL. Recently, a third primase subunit named PriX was identified in the archaeon *Sulfolobus solfataricus*. PriX is essential for primer synthesis and is structurally related to the Fe-S cluster domain of eukaryotic PriL. Here we show that PriX contains a nucleotide-binding site required for primer synthesis, and demonstrate equivalence of nucleotide-binding residues in PriX with eukaryotic PriL residues that are known to be important for primer synthesis. A primase chimera, where PriX is fused to a truncated version of PriL lacking the Fe–S cluster domain retains wild-type levels of primer synthesis. Our evidence shows that PriX has replaced PriL as the subunit that endows primase with the unique ability to initiate nucleic acid synthesis. Importantly, our findings reveal that the Fe–S cluster is not required for primer synthesis.

[1] Department of Biochemistry, University of Cambridge, Tennis Court Road, Cambridge, UK. [2] Molecular and Cellular Biochemistry Department, Indiana University, Bloomington, 47405, USA. [3] Biology Department, Indiana University, Bloomington, 47405, USA. Correspondence and requests for materials should be addressed to S.D.B. (email: stedbell@indiana.edu) or to L.P. (email: lp212@cam.ac.uk)

Initiation of DNA synthesis during DNA replication relies on a specialised DNA-dependent RNA polymerase, known as primase. Primase possesses the unique ability to synthesise de novo short RNA primers that are extended by the replicative DNA polymerases on both strands of the unwound parental DNA[1,2]. The strict requirement for an oligonucleotide primer by DNA polymerases has meant that primases are widespread in all kingdoms of life. Because of the antiparallel nature of the DNA double helix and the obligate 5′-to-3′ direction of nucleotide polymerisation by DNA polymerases, priming events must take place at frequent and regular intervals during lagging-strand synthesis. In practice, leading-strand synthesis might also need re-priming due to occasional interruptions caused by damaged or blocked template. Thus, primase is a core component of the multi-protein assembly known as the replisome that performs DNA synthesis in all organisms[3].

Primases synthesise short RNA oligonucleotides ranging in size between 8 and 12 nucleotides. Primer synthesis comprises the rate-limiting step of initiation, when the initial di-nucleotide linkage is formed, followed by elongation and active termination[4]. After extension by the replicative DNA polymerase, the RNA primer is removed prior to ligation of the Okazaki fragments[5]. In bacteria, primase is a single-chain multi-domain polymerase known as DnaG, which interacts directly with the DnaB helicase at the fork to prime DNA synthesis[6]. In archaea and eukarya, primase is a constitutive heterodimer of a catalytic subunit, PriS or Pri1, and a regulatory subunit, PriL or Pri2[7]. In eukaryotic replication, primase activity is tightly coupled to that of DNA polymerase α (Pol α), to which it is constitutively bound in a heterotetrameric assembly, the Pol α/primase complex[8]. After completion of RNA primer synthesis, Pol α's catalytic subunit performs a limited primer extension with deoxynucleotides to generate an RNA-DNA oligonucleotide that is elongated by Pol δ and Pol ε on the lagging and leading-strand templates, respectively[9].

We now have a detailed structural knowledge of both primase subunits, from crystallographic studies of archaeal and eukaryotic primases[10–13]. PriS adopts a single-domain fold, which is considerably simpler than that of other eukaryotic DNA and RNA polymerases, and is also found in prokaryotic multi-functional polymerases that perform DNA end-joining repair and in plasmid-encoded primases[14,15]. More recently, the PriS fold was identified in PrimPol, a single-subunit enzyme that shares with primase the ability of priming DNA synthesis and has a functional role as translesion polymerase during replicative stress[16–18]. In the PriS fold, the active site is located within an exposed groove in the concave side of two divergent β-sheets, surrounded on the outside by α-helices[10,12,13,19]. PriL in contrast is entirely α-helical, and docks on one side of PriS, providing it with a protruding arm at the end of which is located its most conserved region, an autonomously folded domain termed PriL-CTD[11–13].

Although the mechanistic details of PriL's involvement in primase activity has remained obscure for a long time, eukaryotic PriL is known to be important for primer synthesis[20–22], and its C-terminal domain contains amino acids that are essential for primase activity[20]. Later studies demonstrated that the PriL of archaeal and eukaryotic primases contain an Fe–S cluster within its C-terminal domain, that the integrity of the PriL-CTD is important for primer synthesis and that the PriL-CTD structure bears remarkable similarity to the active-site region of DNA photolyases, hinting at its evolutionary origin as an ancient single-stranded (ss) DNA-binding module[23–25]. More recently, the structure of the human PriL-CTD bound to an RNA/DNA helix has provided a structural basis for the engagement of PriL with the RNA primer during primer synthesis and for the specific recognition of the triphosphate at the 5′-end of the RNA

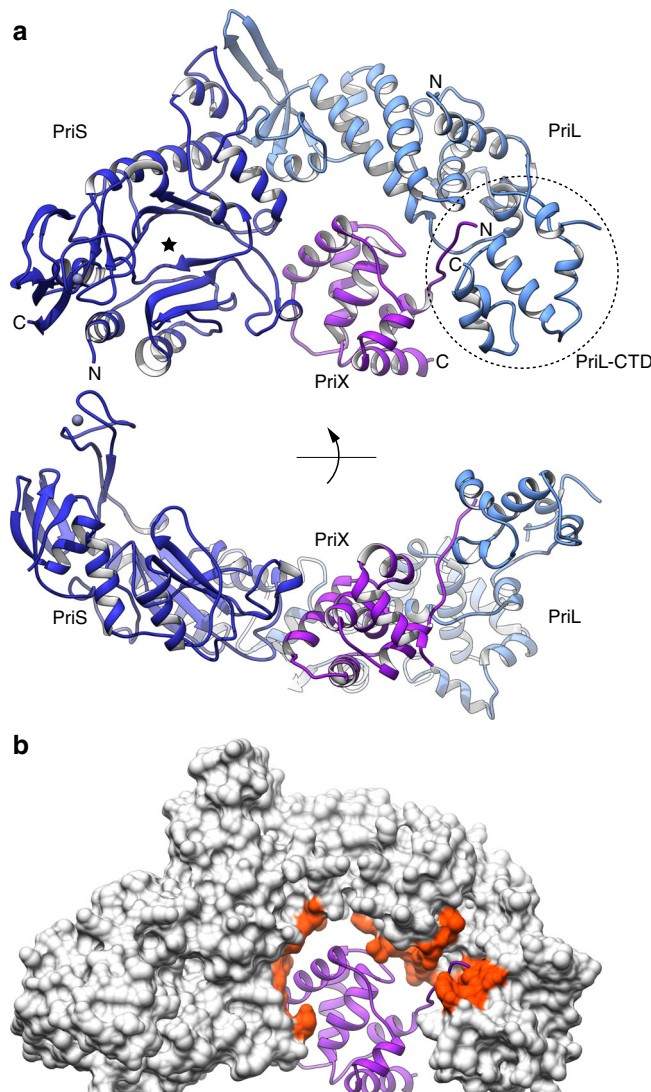

**Fig. 1** Crystal structure of the heterotrimeric primase PriSLX from *S. solfataricus*. **a** Two views of the PriSLX structure. The three subunits are drawn as ribbons, colored blue (PriS), light blue (PriL) and purple (PriX). The position and extent of the PriL-CTD is highlighted by a dashed oval. The N- and C-terminal residues included in the crystallographic model are labeled with N and C, respectively. The known location of the PriS active site is indicated by a star. **b** View of the PriSLX structure that highlights the mode of PriX binding to the PriSL subunits. PriX is shown as a purple ribbon, as in panel A, whereas the PriS and PriL subunits are drawn as a solvent-accessible gray molecular surface. The interface regions of PriS and PriL that are within 5 Å of PriX are colored red

primer[26]. Unexpectedly, a recent report has revealed that the primase of the hyperthermophilic archaeon *Sulfolobus solfataricus* contains a third subunit, named PriX[27]. This archaea-specific subunit is structurally related to the most conserved part of the eukaryotic PriL-CTD[27]. PriX is required for primase activity and the protein is encoded by a gene that is essential for viability[27].

Together, these biochemical and structural findings converge towards a model of primer synthesis in which the critical step of di-nucleotide formation requires the juxtaposition of the nucleotide triphosphate (NTP) bound in the active site of PriS (the elongation site) with a second NTP bound within the PriL-CTD (the initiation site)[7,12], the latter nucleotide eventually occupying the 5′-end position in the primer. Although the

architecture of the elongation site in PriS is now well-characterised[12, 28], the structural basis for PriL's involvement in primer synthesis, via NTP binding and possible functional role of its Fe–S cluster, remains uncertain. Furthermore, the presence of a Fe–S cluster cofactor has raised the important question of whether its redox state plays a functional role in primase activity[29]. Here we provide evidence that addresses these two important questions in primase activity, by identifying the initiation site in the *S. solfataricus* PriSLX primase and demonstrating that its Fe–S cluster is dispensable for primer synthesis.

## Results

**Crystal structure of PriSLX.** To determine a structural basis for the role of PriX in priming DNA synthesis during DNA replication, we determined the X-ray crystal structure of heterotrimeric PriSLX from *S. solfataricus* at 2.9 Å resolution (Fig. 1a and Table 1). In the structure, the PriSL subunits adopt an identical reciprocal arrangement as observed previously in the crystal structure of the truncated PriSL[11]. The PriSLX structure reveals that the six-helix bundle of globular PriX is docked within the concave arch formed by PriS and PriL, and is coplanar with them. The amino-terminal tail of PriX, which was disordered in the crystal structure of the isolated protein[27], extends away from its helical core and reaches the carboxy-terminal domain of PriL (PriL-CTD). The PriL-CTD appears to be only loosely connected to the rest of PriL and is partially disordered, in agreement with previous structural data for human primase[13].

Because of its position near the PriS-PriL interface, PriX makes contacts with both PriS and PriL subunits (Fig. 1b). The extent of the binary interfaces as seen in the crystal structure is modest ($430 \, \text{Å}^2$ with PriS and $660 \, \text{Å}^2$ with PriL). We note that the interaction between PriX and PriL is likely to be more extensive than what observed in our crystallographic model, as biochemical pull-down experiments show that the first 27 amino acids of PriX support binding to PriL-CTD, and that the minimal binding region spans PriX amino acids 10 to 27 (Supplementary Fig. 1). In the structure, the first PriX amino acid that could be modeled in the density map is P48; the PriX region spanning residues 48 to 56 sits in extended conformation in the groove formed by the PriL-CTD and the rest of PriL's larger helical structure. Our inability to observe a larger portion of the PriL-bound PriX region in the density map is likely due to the loose connection of PriL-CTD with the rest of primase, as mentioned previously. Thus, the interaction of the N-terminal tail of PriX with the PriL-CTD is the critical contact that keeps PriX anchored to PriSL, which is in agreement with previous observations[27], and the additional contacts with PriS and PriL are likely to be important for the correct positioning of PriX relative to the rest of primase.

**A nucleotide-binding site in PriX.** PriSLX was crystallized in the presence of RNA/DNA and the non-hydrolysable nucleotide analog AMPCPP. Although no RNA/DNA was detected in the density map, two AMPCPP molecules were clearly visible. The first AMPCPP was bound in the active site of PriS, in what is considered the elongation site of primase[12] (Supplementary Fig. 2A). The nucleotide is bound in the same position and orientation as observed for human and yeast primase[12, 28]. The 2′-hydroxyl group of the ribose in the bound AMPCPP makes two hydrogen bonds to the main-chain carbonyl and nitrogen moieties of PriS L245 and R247, respectively (Supplementary Fig. 2B). A similar set of contacts with the 2′-hydroxyl of the ribose were observed in the structure of human PriS bound to UTP[12] and might help explain the preference of primase for ribo-over deoxy-ribonucleotide polymerisation.

Interestingly, the second AMPCPP located in the density map was bound to PriX, at the edge of the PriX-PriS interface (Fig. 2 and Supplementary Fig. 3). The AMPCPP-binding site of PriX is formed by the N-terminal turn of its second α-helix, at the center of its globular fold, which grips the triphosphate moiety of the AMPCPP in a manner reminiscent of the phosphate-binding loop (P-loop or Walker A motif) of ATP- and GTP-binding proteins[30]. The ribose of protein-bound AMPCPP remains exposed to solvent, whereas the adenine base becomes sandwiched between PriX and PriS. Lack of specific ribose recognition suggests that PriX binding does not discriminate between NTP and dNTP. The side chains of the basic 72-RKR-74 motif in PriX contact the triphosphate, with R72 and R74 buttressed by supporting interactions provided by Y103 and D140, respectively (Fig. 2b). In addition, D70 binds to the triphosphate moiety via the shared coordination of a $Mn^{2+}$ ion.

**Table 1 Data collection and refinement statistics**

|  | PriSLX | Chimera |
|---|---|---|
| *Data collection* | | |
| Wavelength (Å) | 0.97625 | 0.97949 |
| Resolution range (Å) | 46.91–2.906 | 46.17–3.005 |
|  | (3.01–2.906) | (3.113–3.005) |
| Space group | P 4₁ | C 1 2 1 |
| Unit cell | 104.9 Å 104.9 Å | 205.1 Å 36.3 Å 110.9 Å |
|  | 229.8 Å 90 90 90 | 90 96.907 90 |
| Total reflections | 504421 (48565) | 53604 (5064) |
| Unique reflections | 54380 (5392) | 16582 (1556) |
| Multiplicity | 9.3 (9.0) | 3.2 (3.3) |
| Completeness (%) | 99.96 (99.93) | 98.63 (95.17) |
| Mean I/sigma(I) | 15.09 (1.47) | 9.18 (1.86) |
| Wilson B-factor | 89.45 | 77.00 |
| R-merge | 0.105 (1.404) | 0.1168 (0.8095) |
| R-meas | 0.1112 (1.489) | 0.1402 (0.9655) |
| R-pim | 0.03641 (0.4952) | 0.07654 (0.5204) |
| CC1/2 | 0.998 (0.527) | 0.99 (0.535) |
| *Refinement* | | |
| Reflections used in refinement | 54374 (5392) | 16579 (1556) |
| Reflections used for R-free | 2677 (271) | 828 (75) |
| R-work | 0.2283 (0.3081) | 0.2412 (0.3726) |
| R-free | 0.2713 (0.3434) | 0.2980 (0.3845) |
| CC(work) | 0.867 (0.686) | 0.929 (0.582) |
| CC(free) | 0.919 (0.582) | 0.924 (0.432) |
| Number of non-hydrogen atoms | 11542 | 5117 |
|   Macromolecules | 11412 | 5109 |
|   Ligands | 130 | 1 |
|   Solvent |  | 7 |
| Protein residues | 1405 | 625 |
| RMS(bonds) | 0.002 | 0.001 |
| RMS(angles) | 0.54 | 0.38 |
| Ramachandran favored (%) | 93.47 | 93.85 |
| Ramachandran allowed (%) | 6.09 | 5.50 |
| Ramachandran outliers (%) | 0.44 | 0.65 |
| Rotamer outliers (%) | 3.29 | 3.68 |
| Clashscore | 6.57 | 4.36 |
| Average B-factor | 101.48 | 70.23 |
|   Macromolecules | 101.53 | 70.26 |
|   Ligands | 97.02 | 65.74 |
|   Solvent |  | 51.67 |

Statistics for the highest-resolution shell are shown in parentheses

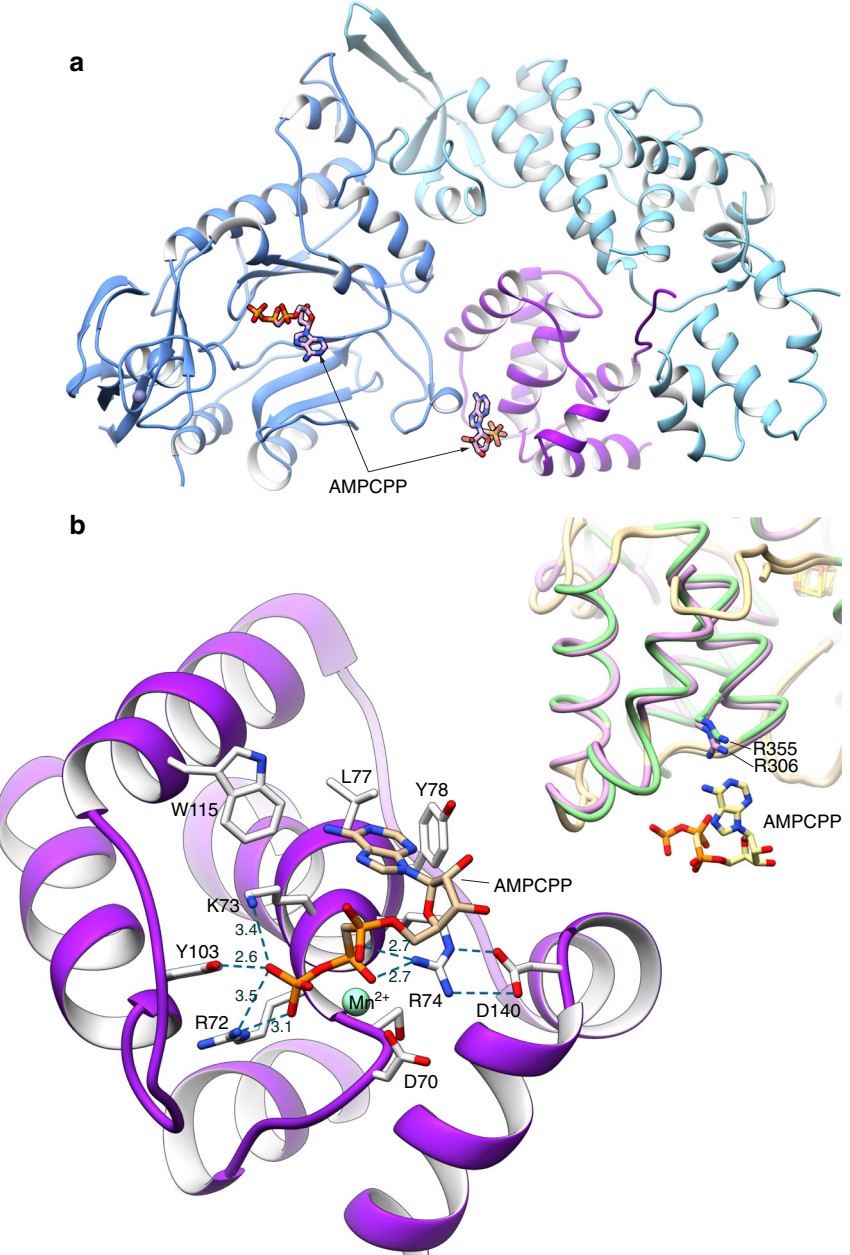

**Fig. 2** The nucleotide-binding site in PriX. **a** View of the PriSLX structure, highlighting the position of the two bound AMPCPP molecules. The primase structure is drawn as in Fig. 1a, the AMPCPP is shown in stick representation. **b** Close-up view of the NTP-binding site of PriX with bound AMPCPP. The side chains of the amino acids that form the NTP-binding site are shown. Polar contacts between side chains of NTP-binding residues and AMPCPP are drawn as dashed lines, together with their distances in Ångström. The top-right inset shows a superposition of the yeast (green; PDB ID 3LGB) and human (pink; PDB ID 3Q36) PriL-CTD structures. The side chain of arginine residues R355 (yeast) and R306 (human), that are essential for initiation of primer synthesis, are also shown. The AMPCPP molecule of superimposed PriX (protein not shown) is overlaid on the PriL-CTD structures, to highlight its proximity to R355 and R306

The observation of a PriX-bound AMPCPP in the crystal structure of PriSLX suggests the presence of a functional nucleotide-binding site in PriX. Multiple sequence alignment of PriX sequences shows that D70, R72 and R74 are invariant residues (Supplementary Fig. 4A), as would be expected for amino acids with an essential functional role in primase activity. Intriguingly, superposition of PriX with eukaryotic PriL-CTD, based on their known three-dimensional similarity[27], shows that structurally equivalent amino acids R355 of yeast PriL and R306 of human PriL superimpose with PriX R72 (Fig. 2b and Supplementary Fig. 4B). Importantly, human R306 and yeast

R355 in PriL have been shown to be essential for the initiation step of primer synthesis[12, 20, 23]. Our observation indicates that these PriL residues could contribute to an analogous NTP-binding site in the eukaryotic PriL-CTD, thus potentially explaining their role in primase activity.

To test a possible role of the NTP-binding site in PriX, we designed three single-point alanine mutants in PriX: D70, R72, R74 (Fig. 2b), and analyzed their enzymatic activity in primer synthesis assays (Fig. 3a, b), using a range of conditions, including different template DNA sequences, primase concentrations, and reaction conditions. Priming reactions on poly(dT) ssDNA

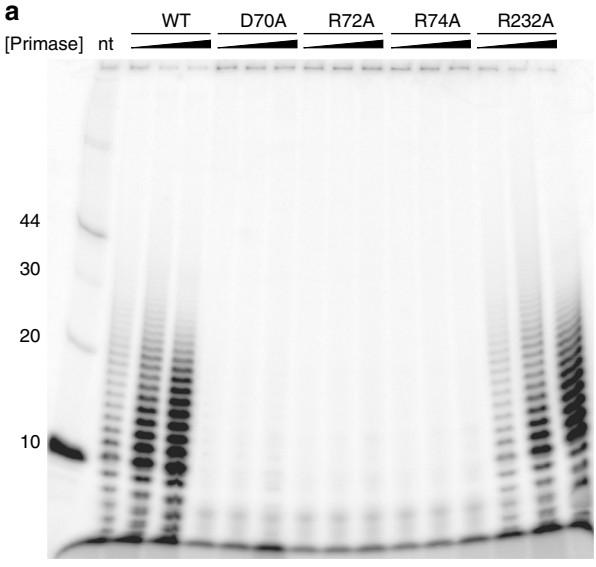

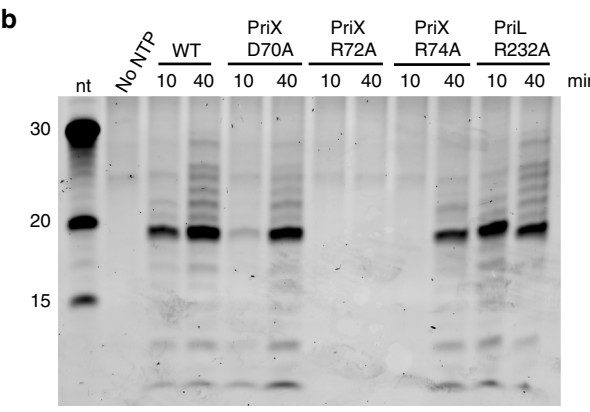

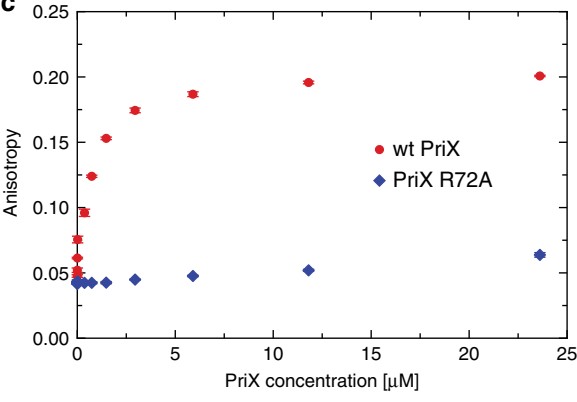

**Fig. 3** PriX NTP-binding residues are necessary for RNA primer synthesis. **a** Primer synthesis assay on poly(dT) ssDNA for increasing concentrations (125 nM, 250 nM and 500 nM) of wild-type and mutant PriSLX. The reaction products were separated by PAGE under denaturing conditions and detected by phosphorimaging of the incorporated $^{32}$P-αATP. **b** Primer synthesis assay on mixed-sequence ssDNA 100mer with fixed concentration of wild-type and mutant primase, at 10′ and 40′ time points. The reaction products were separated by denaturing PAGE, stained with SYBR Gold and detected by phosphorimaging. **c** ATP binding by wild-type and R72A PriX. Binding was measured by fluorescence polarization of fluorescein-labeled ATP in the presence of increasing amounts of primase

showed no activity for any the mutants (Fig. 3a); on a mixed-sequence DNA template, R72A mutant showed no activity, whereas D70A and R74 mutants yielded some primer products after prolonged incubation (Fig. 3b). As a control, alanine mutation of PriL R232, which is close to R306 and R355 of human and yeast PriL, based on sequence conservation of archaeal/eukaryotic primases[23], had no impact on primase activity. Furthermore, measuring fluorescein-ATP binding to isolated PriX using fluorescence polarisation showed that PriX binds ATP with sub-micromolar affinity and that the R72A mutation abolishes binding (Fig. 3c). Together, these results indicate that conserved PriX amino acids that bind to AMPCPP in the crystal structure of PriSLX are essential for primer synthesis, thus identifying the NTP-binding site in PriX as the initiation site of primase.

**Functional replacement of PriL-CTD by PriX in primer synthesis.** The identification of an NTP-binding site in PriX that is important for primer synthesis by archaeal primase suggested a functional replacement of the PriL-CTD, which is essential for initiation of primer synthesis in the eukaryotic primase, by PriX. To test this hypothesis, we designed and prepared a heterodimeric version of the *S. solfataricus* primase (named Chimera), which contained wild-type PriS and a second subunit where amino acids 1 to 211 of PriL were fused to amino acids 42 to 154 of PriX, thus excluding the PriL-CTD (Fig. 4a). Strikingly, Chimera was fully active in primer synthesis assays on both poly(dT) and mixed-sequence DNA templates, in support of the hypothesis that PriX had replaced PriL-CTD as the functional module that cooperates with PriS in *S. solfataricus* primase (Fig. 4b, c). Taken together, these results show that PriX is necessary and sufficient to endow primase with the ability to initiate RNA primer synthesis.

To analyze further the biochemical properties of Chimera and of PriSLX bearing single-point PriX mutations D70A, R72A and R74A, we analyzed their ability to support DNA synthesis using a primase-dependent reconstituted DNA replication assay with purified *S. solfataricus* protein components and an M13 ssDNA template (Supplementary Fig. 5). The experiments showed that Chimera was able to prime levels of DNA synthesis that were comparable to those of wild-type PriSLX (Fig. 5), thus confirming the results of the primase assay (Fig. 4). Furthermore, the R72A and R74A mutants showed barely detectable and greatly diminished levels of DNA synthesis, respectively; the D70A mutant also showed clearly reduced DNA synthesis.

**PriX is required for the initiation of primer synthesis.** To investigate further the functional role played by PriX in primase activity, we examined the nature of the impairment in primer synthesis caused by the PriX mutants D70A, R72A and R74A. Primer synthesis on poly(dT) using $^{32}$P-αATP showed that the defect of the PriX mutants was specific to the initiation step, as clearly detectable extension products were observed when an oligo(A$_8$) primer was provided in the reaction (Fig. 6a). To be able to compare the ability of the PriX mutants to extend an existing primer relative to the wild-type primase, the extension assay was repeated using $^{32}$P-labeled oligo(A$_8$). The results of the experiment showed that PriX mutants had comparable levels of activity to wild-type primase, and that both PriX mutants and Chimera were able to synthesise longer extension products (Fig. 6b).

These data proved that, when presented with both a DNA template and an RNA primer, the defect in NTP binding by PriX did not affect the ability of primase to function as DNA-dependent RNA polymerase. Thus, PriX has a specific, essential role in the initiation step of primer synthesis. As formation of the

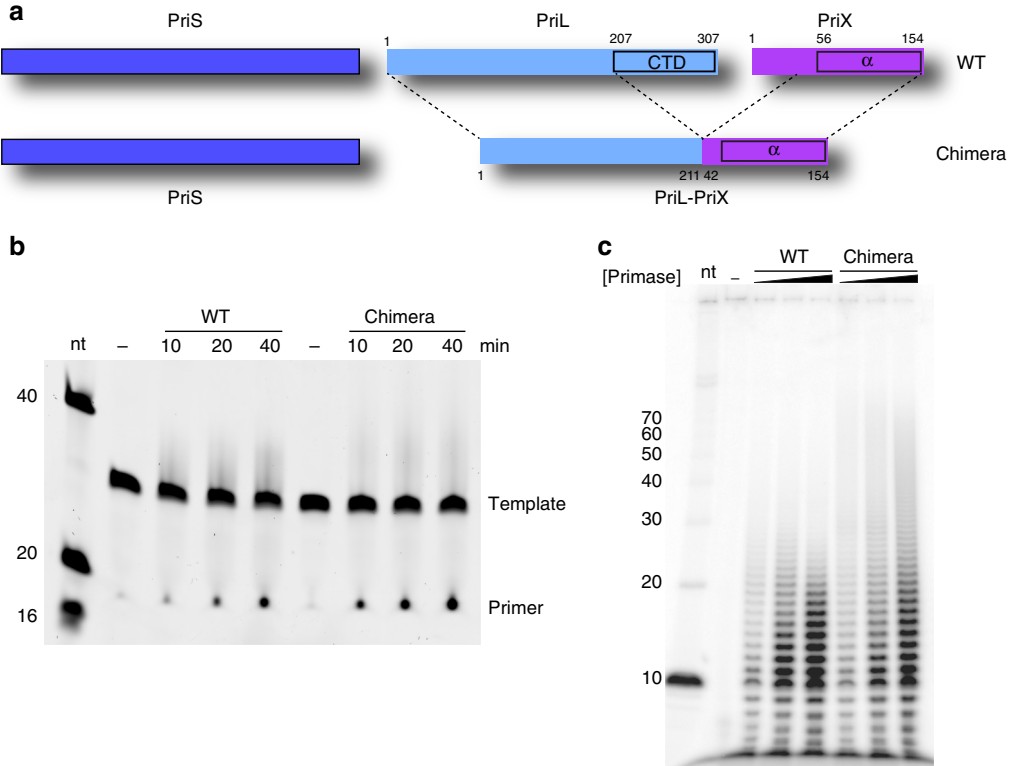

**Fig. 4** Design and functional characterization of Chimera. **a** Subunit structure of wild-type PriSLX and Chimera. In the second subunit of Chimera, amino acids 1 to 211 of PriL are fused to amino acids 42 to 154 of PriX. **b** Primer synthesis assay on mixed-sequence ssDNA 30mer with fixed concentration of wild-type and mutant primase, at 10′, 20′ and 40′ time points. The reaction products were separated by denaturing PAGE, stained with SYBR Gold and detected by phosphorimaging. **c** Primer synthesis assay on poly(dT) ssDNA with increasing concentrations (125 nM, 250 nM, and 500 nM) of wild-type PriSLX and Chimera. The reaction products were separated by PAGE under denaturing conditions and detected by phosphorimaging of the incorporated [32]P-αATP

first phosphodiester bond between two nucleotides is the critical catalytic step in the initiation step of primer synthesis, we sought to determine whether provision of the ribo-di-nucleotide ApA would suffice to compensate for the initiation defect of the PriX mutants. Indeed, the PriX mutants were able to extend the ApA dinucleotide, albeit by a single nucleotide (Fig. 6c). The reason for the limited extension might be the inefficient interaction of primase with such as short primer, when operating in polymerase mode.

**The crystal structure of PriSLX is active in primer elongation**. The position of PriX and its bound nucleotide relative to the PriS active site is not immediately suggestive of a possible mechanism explaining PriX involvement in primer synthesis. We set out to ascertain in the first instance whether the relative arrangement of primase subunits observed in the crystal structure of PriSLX is present in solution. To this purpose, we adopted a structure-guided site-specific cross-linking approach: we chose pairs of residues that are present at the PriS–PriX interface in the PriSLX crystal structure (Fig. 7a), and selected them for cysteine mutagenesis. The primase containing the two cysteine mutations was then subjected to cross-linking using the cross-linker bis-mal-eimidoethane (BMOE), and the pattern of cross-linked primase subunits was analyzed by SDS-PAGE.

The results of the cross-linking analysis for the amino acid pair PriS E34 and PriX K110, as well as for control primase mutants containing either PriS T11C, PriS E34C or PriS T11C, PriX K110C mutations, where T11 is located in the disordered N-tail of PriS, are reported in Fig. 7b. Whereas the wild-type PriSLX was

not affected by the presence of BMOE, and the control experiment showed a ladder of products indicative of non-specific cross-linking, the PriS E34C, PriX K110C double mutant primase showed clearly a predominant product corresponding to cross-linked PriS and PriX. This result indicates that the crystallographic model determined for the PriSLX crystals does exist and it is prevalent in solution. The cross-linking experiment was performed for both PriSLX and Chimera, yielding equivalent results, in further support of Chimera's recapitulation of the behavior of wild-type PriSLX.

Interestingly, when the cross-linking was repeated with primase bound to an RNA/DNA molecule, a different result was obtained according to whether the RNA contained a triphosphate group on its 5′-nucleotide (5′-pppRNA) (Supplementary Fig. 6). Thus, whereas addition of RNA/DNA to the reaction did not change the amount of cross-linked PriS-PriX subunits, the presence of 5′-pppRNA/DNA reduced noticeably the amount of cross-linked material. These results can be interpreted to indicate that engagement with the templated 5′-pppRNA caused a rearrangement in primase structure, away from the conformation observed in the PriSLX crystals.

Having ascertained that the primase structure of the PriSLX crystals does exist in solution, we aimed to determine whether it is functional in primer synthesis. In order to do this, primase assays were performed with a PriS E34C, PriX K110C double mutant that had been cross-linked and purified by gel-filtration chromatography (Supplementary Fig. 7). As it proved to be easier to cross-link the Chimera double mutant to completion, it was used in the primase assays. Primer initiation assay showed no primer synthesis by the cross-linked Chimera (Fig. 8a), in support

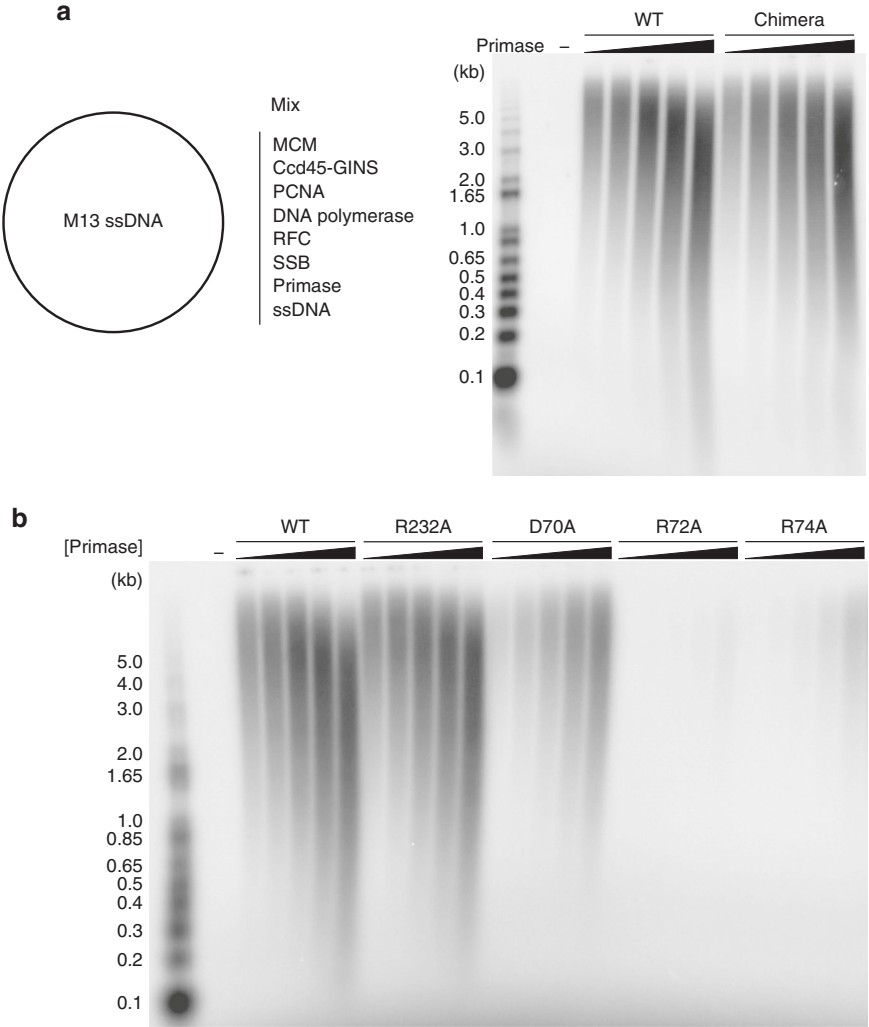

**Fig. 5** DNA replication assays. Relative levels of DNA synthesis were measured in a reconstituted DNA replication assay with purified proteins on a M13 ssDNA template, in the presence of wild-type primase (control) and Chimera (**a**) or wild-type primase (control) and single-point mutants R232A, D70A, R72A and R74A (**b**). The reactions contain 3.125 nM, 6.25 nM, 12.5 nM, 25 nM, or 50 nM primases. The reaction products were separated by alkaline agarose gel electrophoresis and detected by phosphorimaging of the incorporated $^{32}$P-αdCTP

of the view that the PriSLX conformation, as seen in the crystal structure, is not compatible with primer initiation. However, RNA primer extension assays using either NTP (Fig. 8b) or dNTP (Fig. 8c) did show primer elongation, indicating that the crystal structure of PriSLX represents a biochemically active and potentially physiologically relevant conformation.

## Discussion

Here we have presented the crystal structure of the heterotrimeric PriSLX primase from *S. solfataricus* and have clarified the role of its novel PriX subunit in RNA primer synthesis. We have shown that PriX has replaced the Fe–S cluster domain of PriL as the primase module responsible for endowing primase with the ability to initiate nucleic acid synthesis. We have further shown that the critical role of PriX in primer initiation stems from its ability to bind NTPs and have identified the amino acids involved.

The nucleotide-binding site of PriX bears resemblance to the phosphate-binding loop (P-loop or Walker A motif) of ATP- and GTP-binding proteins. The β- and γ-phosphates of the bound nucleotide are nested within the N-terminal turn of the central α-helix in the PriX globular fold, and make hydrogen bonds with the main-chain nitrogens of the 72-RKR-74 motif. Furthermore,

the three phosphates are bound to a $Mn^{2+}$ ion, with D70, providing an additional point of coordination. Alanine mutation of D70, R72, R74 abolish or severely impair the initiation step of primer synthesis, but leave unaffected the ability of primase to extend an existing primer, indicating that the NTP-binding site in PriX represents the initiation site of primase.

The presence of a novel NTP-binding site in PriX, distinct from the elongation site in PriS, provides support to the model that archaeal/eukaryotic primases possess two independent NTP-binding sites, the elongation and initiation sites, that must become transiently juxtaposed to promote synthesis of the first di-nucleotide in the RNA primer (Fig. 9). The conformation change required to juxtapose the first two nucleotides highlights a distinctive mechanism of initial dinucleotide synthesis for primase. We speculate that the initiation-competent conformation of primase may align the first two nucleotides for phosphodiester bond formation in a manner similar to that suggested for Prim-Pol[18]. The proposed primase initiation mechanism is different from the initiation mechanism employed by transcription RNA polymerases, where the active site is buried deep within the multi-subunit enzyme and can accommodate two NTPs during transcription initiation[31]. The structural basis for the conformational rearrangements that allow archaeal and eukaryotic primases to

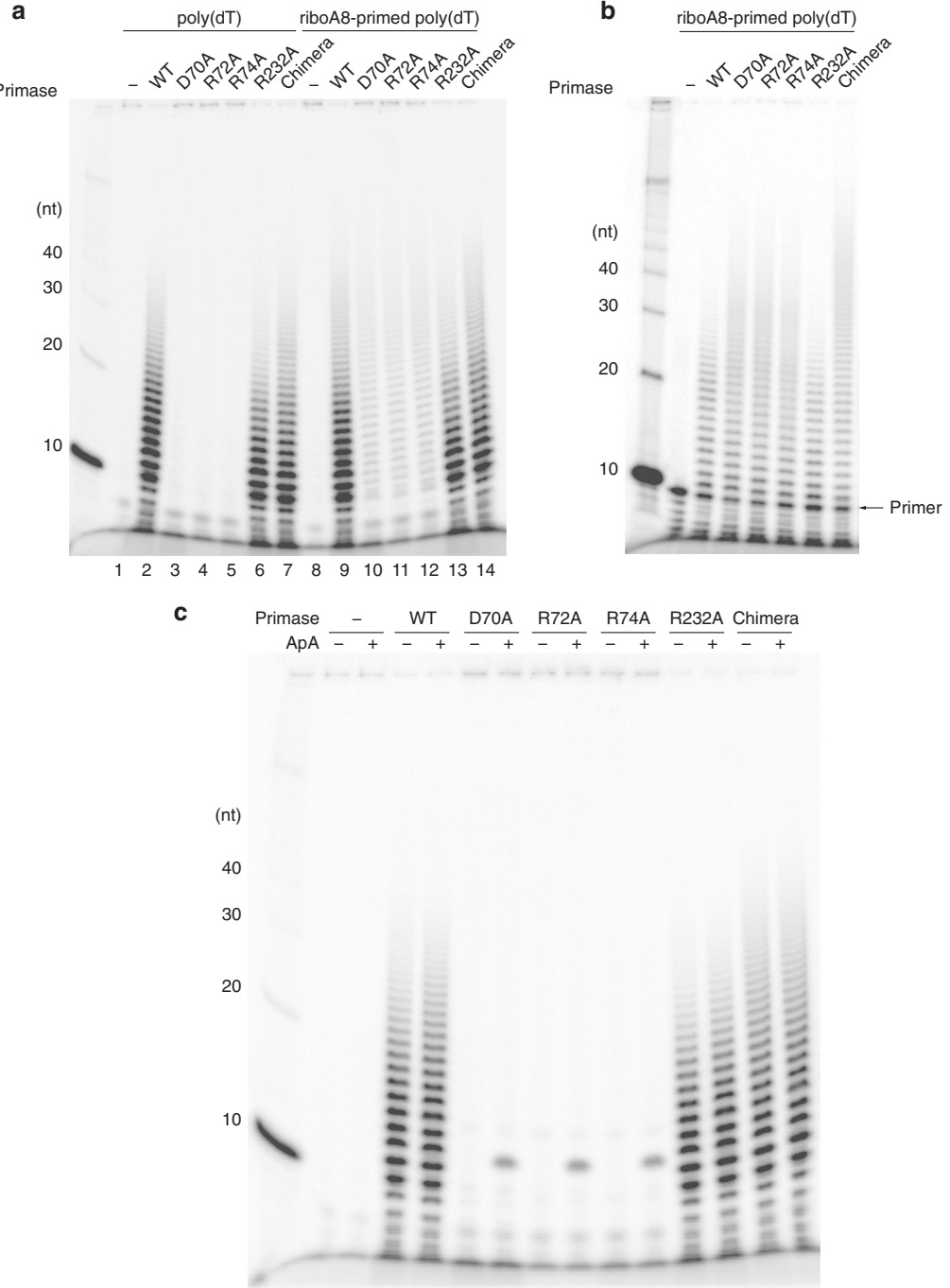

**Fig. 6** The PriX mutants are defective in primer initiation but competent in primer elongation. All the reactions contain 500 nM primase. **a** Primer synthesis assay for wild-type, chimera and mutant primase, in the presence of either poly(dT) ssDNA template (lanes 2–7) or poly(dT) ssDNA template and ribo (A$_8$) primer (lanes 9–14). The reaction products were separated by PAGE under denaturing conditions and detected by phosphorimaging of the incorporated $^{32}$P-αATP. **b** Primer extension assay for wild-type, chimera and mutant primase, in the presence of a poly(dT) ssDNA template and 5'-labeled $^{32}$P-ribo(A$_8$) primer. The reaction products were separated by PAGE under denaturing conditions and detected by phosphorimaging. **c** Primer synthesis assay for wild-type, chimera and mutant primase, in the presence of a poly(dT) ssDNA template and ApA di-nucleotide. The reaction products were separated by PAGE under denaturing conditions and detected by phosphorimaging of the incorporated $^{32}$P-αATP. Note that short oligo-ribonucleotides have been demonstrated to have aberrantly slow mobility on denaturing polyacrylamide gels[40]

achieve the initiation state competent for di-nucleotide synthesis remains currently unknown.

The identification of a second NTP-binding site in primase was made during attempts to co-crystallize primase with RNA/DNA and a non-hydrolysable nucleotide analog, AMPCPP. The conformation of PriSLX observed in the crystals appears to be predominant in solution, as judged by the cross-linking experiments

of Fig. 7b, and it is a conformation that is also adopted by the Chimera. The ability of the cross-linked Chimera to extend an existing primer, when constrained in a similar conformation to the PriSLX crystal structure, argues that, in addition to existing in solution, the conformation is physiologically relevant. The fact that the cross-linked Chimera was exclusively active in the extension, but not initiation, of an RNA primer, points to the

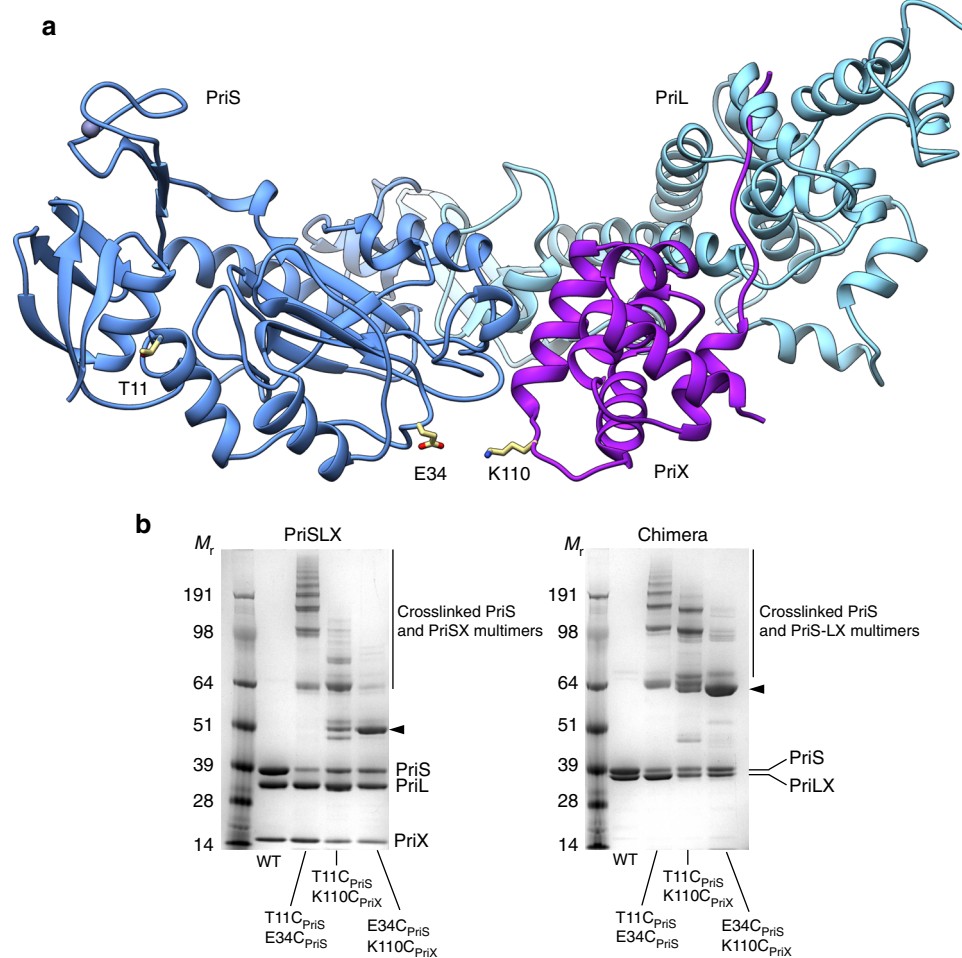

**Fig. 7** Structure-based cross-linking of PriSLX. **a** Ribbon diagram of PriSLX. The side chains of PriS E34 and PriX K110, that were targeted for cysteine mutagenesis, are drawn in stick representation at the PriS–PriX interface. The side chain of PriS T11, that was used as control in the cross-linking reactions, is also shown. **b** BMOE cross-linking of PriSLX and Chimera PriS E34C, PriX K110C double mutants. The PriS T11C, E34C and PriS T11C, PriX K110C double mutants were used as controls. The products of the cross-linking reactions were separated by SDS-PAGE and stained with Coomassie Blue. The arrowheads mark the product of the specific cross-linking between PriS E34 and PriX K110

conformation of the crystal structure as representing a 'polymerase' mode. From these observations, it follows that PriX must transition between different conformational states to be able to initiate primer synthesis. Interestingly, the crystal structure of Chimera shows that covalently-linked PriX occupies a similar but not identical position relative to PriX in the PriSLX structure, possibly representing an intermediate point in a putative PriX trajectory between initiation and extension states (Supplementary Fig. 8).

Previous studies had shown that the PriL subunit provides the eukaryotic primase with the ability to initiate primer synthesis, by means of a conserved C-terminal Fe–S cluster domain, the PriL-CTD[23]. A surprising implication of our findings is that PriX has replaced PriL as the primase subunit that provides this essential initiation ability in the archaeal primase of *S. solfataricus* (Fig. 9). Thus, our work extends significantly the earlier demonstration of structural similarity between archaeal PriX and eukaryotic PriL-CTD[27], by showing that the similarity includes their functional roles. We prove this point beyond doubt, by demonstrating that an engineered version of PriSLX, lacking the PriL-CTD retains priming activity.

Furthermore, our findings have important implications for the debated role of the Fe–S cluster in primer synthesis. Recent reports had suggested that the redox activity of the Fe–S cluster of primase is important for primer synthesis, by regulating the DNA-binding affinity of primase[29]. Our evidence of a functional replacement of PriL-CTD by PriX indicates that neither the presence of a Fe–S cluster nor, therefore, its putative redox activity are required for primer synthesis in the archaeal primase of *S. solfataricus*. While it remains possible that eukaryotic primases might operate via a different mechanism of primer synthesis, their close structural and functional relationship with archaeal primases makes it unlikely that the Fe–S cluster would have a redox function in the priming reaction of the eukaryotic enzyme.

## Methods

**Expression and purification of PriSLX and Chimera**. A pETDuet vector expressing PriL, PriS and a pRSFDuet vector expressing PriX were co-transformed into Rosetta2(DE3) cells (Novagen). The resulting colonies were used to inoculate Turbo Broth™ (Molecular Dimensions) media supplemented with 35 µg ml⁻¹ kanamycin, 35 µg ml⁻¹ chloramphenicol, 100 µg ml⁻¹ ampicillin. The culture was grown to an $OD_{600}$ of 1.0 at 37 °C and constant shaking at 200 r.p.m., induced with 1 mM IPTG and incubated at 37 °C and constant shaking for 3 h. The bacteria were collected by centrifugation and the pellets were resuspended in 15 mL of lysis buffer (25 mM Hepes pH 7.2, 150 mM NaCl, 5 mM DTT, EDTA-free protease inhibitor cocktail tablets (Sigma) and 25 units of benzonase (Millipore) per liter of cell culture. The cells were lysed by sonication (Sonics Vibra-Cell™; 5 s pulse and 10 s off, 60% amplitude, 3.5 min). The cell lysate was split into 25 mL aliquots, transferred into 50 mL falcon tubes and incubated in a 70 °C water bath for 20 min.

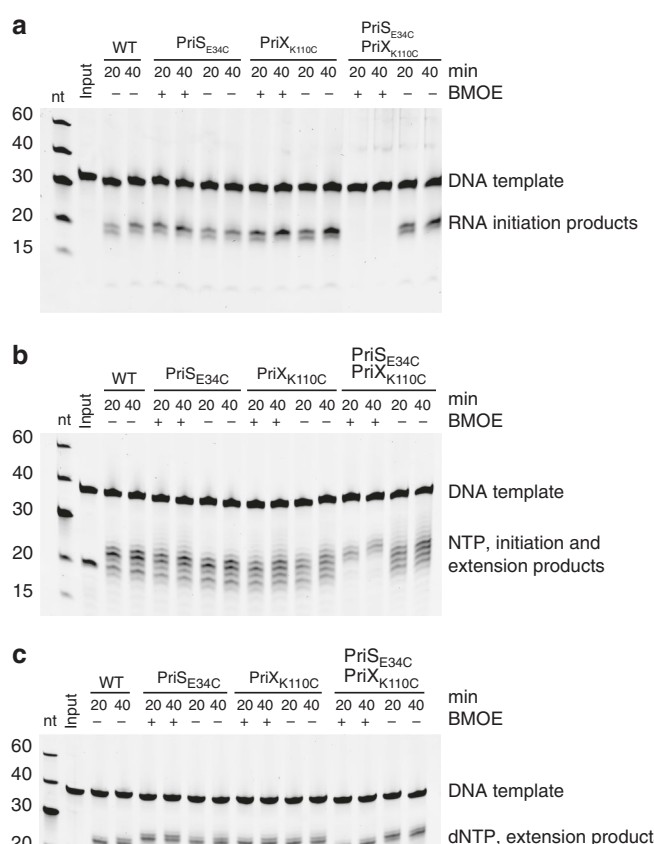

**Fig. 8** Primer synthesis assay for the cross-linked Chimera. For all experiments, wild-type Chimera, PriS E34C and PriX K110C single mutants and PriS E34C, PriX K110C double mutants were analyzed. Cross-linking reactions were analyzed at 20′ and 40′ time points, with and without BMOE. The reaction products were separated by denaturing PAGE, stained with SYBR Gold and detected by phosphorimaging. **a** Primer synthesis assay on a mixed-sequence ssDNA 30mer template and NTPs. **b** Primer synthesis assay on a mixed-sequence ssDNA 38mer template, RNA 18mer primer and NTPs. **c** Primer synthesis assay on a mixed-sequence ssDNA 38mer template, RNA 18mer primer and dNTPs

After heat denaturation, the cell suspension was centrifuged at 18,000×g for 1 h. The supernatant was filtered through a 5.0 μm syringe filter and the filtrate loaded onto a 5 mL Heparin HP column (GE Healthcare). The Heparin column was connected to an ÄKTA purifier system (GE Healthcare) and the protein eluted with a 10-column volume gradient from 150 to 1000 mM NaCl in the presence of 25 mM Hepes 7.2. The pooled fractions were concentrated, loaded onto a Superdex 200™ 16/600 size-exclusion chromatography column (GE Healthcare) and eluted in a 25 mM Hepes pH 7.2 and 150 mM NaCl buffer. Peak fractions corresponding to purified PriSLX were pooled, concentrated and kept in aliquots at −80 °C.

To prepare the Chimera construct, we generated by overlap extension PCR an open-reading frame encoding PriX amino acids 41–154 fused to amino acids 1–211 of PriL, resulting in a PriLX protein lacking the C-terminal domain of PriL (residues 212–307). The construct was cloned into the pETDuet PriSL vector, replacing PriL, for co-expression with the pRSFDuet vector expressing PriX. The resulting PriSLX protein (Chimera) was expressed and purified following the same protocol as wild-type PriSLX.

**Expression and purification of *S. solfataricus* proteins**. Proteins were expressed and purified essentially as previously described: PCNA[32], RFC[29], PolB1-HE[30], SSB[33]. *S. solfataricus* MCM, GINS and Cdc45 were expressed and purified following the scheme described in Xu et al[34].

**Crystallization and structure determination of PriSLX**. PriSLX was crystallized using vapour diffusion in sitting drops at 19 °C, by mixing 200 nL of PriSLX sample with 100 nL of reservoir solution with the mosquito robot (TTP Labtech) in MRC 2 Well sitting drop crystallization plates. The PriSLX sample contained 25 mM Hepes pH 7.2, 150 mM NaCl, 1 mM TCEP, 1 mM MnCl$_2$, 2 mM AMPCPP, 100 μM PriSLX and 120 μM of the RNA-DNA hybrid sequence 5′-(TTCGATCAGG-TAGC)$_{DNA}$(UUUGCUACCUGAUCG)$_{RNA}$-3′. The reservoir solution contained 11% PEG 8000, 21.5% ethylene glycol, 0.1 M bicine/Trizma base pH 8.5, 0.03 M di-, tri-, tetra- and penta-ethyleneglycol (Morpheus crystallization screen, Molecular Dimensions). The crystals were improved by streak seeding into the initial crystallization conditions, and cryoprotected before freezing in liquid nitrogen by addition of reservoir solution to the crystal drop.

X-ray diffraction data of the PriSLX crystals were collected at the I03 beam line of the Diamond Synchrotron Light Source. 1200 frames were collected with an oscillation range of 0.2 degrees and at a wavelength of 0.97625 Å. The data set was indexed in P4$_1$ with unit cell dimensions 104.9 Å, 104.9 Å, 229.8 Å, 90.0°, 90.0°, 90.0°, scaled and integrated with the XDS software package[35]. The structure was solved by Molecular Replacement with Phaser as implemented in Phenix, using the heterodimeric structure of the core primase as input model (PDB ID: 1ZT2)[11, 36, 37]. The quality of the resulting electron density map was improved with Phenix Refine and the available PriX structure (PDB ID: 4WYH) was manually fitted into the density map using Coot[27, 36, 38]. Subsequently, the structure, containing two copies of the PriSLX heterotrimer in the asymmetric unit, was completed with iterative cycles of manual structure building in Coot and refinements in Phenix-Refine to 2.9 Å. The figures used to present the crystal structure were generated with Chimera[39].

**Crystallization and structure determination of Chimera**. The Chimera was crystallized by vapour diffusion at 19 °C. 2 μL of protein solution was mixed with 2 μL of reservoir solution in a 6 well hanging-drop crystallization plate containing 1 mL reservoir solution. The protein was at 60 μM concentration in 25 mM Hepes pH 7.2, 150 mM NaCl, 1 mM TCEP. The reservoir solution consisted of 12% PEG 3350, 200 mM CaCl$_2$, 0.5% n-octyl-β-D-glucoside. For cryo-protection, the mother liquor was replaced by 4 consecutive exchanges of 2 μL cryo-solution (10% PEG3350, 100 mM CaCl$_2$, 100 mM NaCl, 25 mM Hepes pH 7.2, 25% ethylene glycol, 1 mM TCEP). After the buffer exchanges, the crystals were mounted onto nylon loops and frozen in liquid nitrogen.

The X-ray diffraction data were collected at the I02 beam line of the Diamond Synchrotron Light Source. The wavelength was set to 0.97949 Å and 1800 frames were collected with an oscillation range of 0.1 degrees. The data set was processed in the space group C2, with unit cell dimensions 205.1 Å, 36.3 Å, 110.9 Å, 90°, 96.9°, 90° and one copy of Chimera in the asynmmetric unit, using XDS[35]. The phases were obtained by molecular replacement using Phaser[37] and the heterodimeric structure of the core primase as input model (PDB ID: 1ZT2). The structure was refined to 3.0 Å with iterative cycles of manual structure building in Coot[38] and refinements in PhenixRefine[36].

**Pull-down experiments**. The flag-tagged PriX N-terminal constructs were co-expressed with His-Sumo tagged PriL$_{CTD}$ from a pRSF-Duet vector in Rosetta2 DE3 cells (Novagen). Colonies were inoculated into 10 mL of Turbo Broth™ (Molecular Dimensions) with 35 μg mL$^{-1}$ kanamycin and 35 μg ml$^{-1}$ chlor-amphenicol in 50 mL conical tubes and the cultures grown with shaking to an OD$_{600}$ of 1.0, when the temperature was reduced to 20 °C and the cultures induced with 0.3 mM IPTG. After 16 h, the cells were collected by centrifugation at 4000×g for 10 min. 2 mL of 150 mM NaCl, 25 mM Hepes pH 7.2 supplemented with EDTA-free protease inhibitor cocktail tablets and 10 units benzonase was used to resuspend the cell pellets. The cell suspension was transferred into 2 mL tubes and the cells lysed by sonication (30 s, 10 s on and 10 s off, 60% amplitude). The lysate was centrifuged at 4 °C at 17′000×g for 30 min. Subsequently, the lysate was applied onto 40 μL anti-FLAG M2 beads (Sigma-Aldrich) and incubated on a roller-shaker at 4 °C for 1 h. Using a magnetic rack the beads were washed with 3 × 1 mL of 150 mM NaCl, 25 mM Hepes pH 7.2. Finally the protein was incubated for 30 min on ice with 40 μL wash buffer supplemented with 300 μg mL$^{-1}$ triple-FLAG peptide (Sigma-Aldrich) for elution. The eluted samples were mixed with loading buffer and analyzed by SDS-PAGE.

**Mixed-sequence primase activity assays**. The assay reactions were set up by mixing 0.5 μM primase, 0.5 μM template oligonucleotide (for initiation: DNA template 5′-GTTGTCCATTATGTCCTACCTCGTGCTCCT-3′; for extension: DNA template 5′-TTTTTTTTTTTTTTTTTTTTTTCCAGAGAGCGCCCAAACG-3′ and RNA primer 5′-GGUCUCUCGCGGGUUUGC-3′), 1 mM NTP (or dNTP) in the assay buffer: 25 mM Hepes pH 7.2, 100–150 mM NaCl, 5 mM MgCl$_2$ (or MnCl$_2$), 1 mM TCEP. The reactions were incubated in a PCR thermocycler at 60 °C. For each time point, a separate reaction was run and stopped after 10, 20 or 40 min with a 1:1 ratio of reaction mixture to quenching buffer (95% formamide, 25 mM EDTA, xylene cyanole). The samples were then analyzed on denaturing 20% polyacrylamide-urea gel, by electrophoresis in 0.5× TBE buffer at 500 V for 90 min. For visualisation of RNA and DNA, the gel was stained in 0.5x TBE with 1/10′ 000 SYBR Gold (ThermoFisher) for 20 min and imaged using a Typhoon FLA 9500 (GE Healtcare).

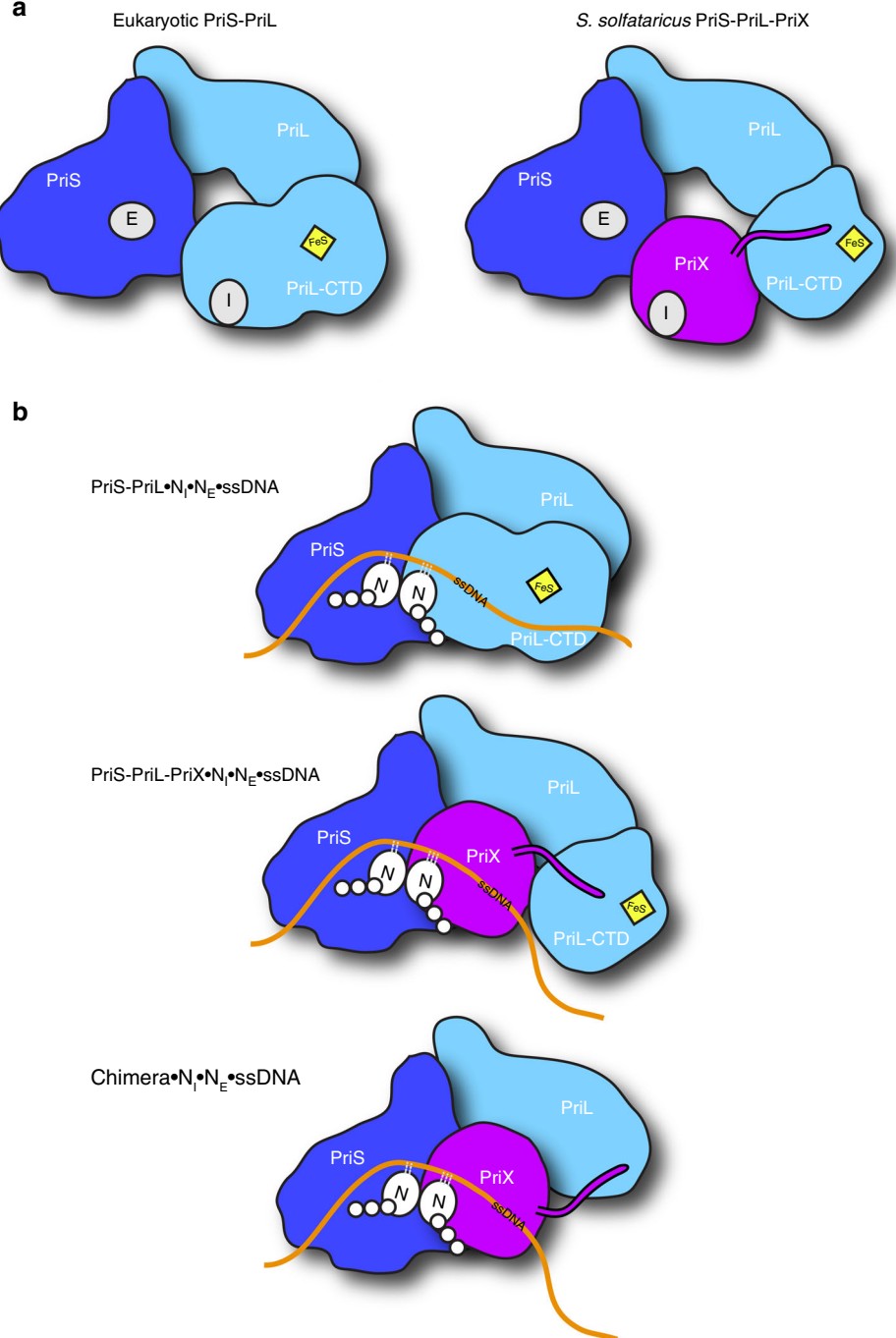

**Fig. 9** Model of RNA primer synthesis by eukaryotic PriSL and archaeal PriSLX, highlighting the respective roles of PriX and PriL-CTD in primer synthesis. **a** Subunit structure of eukaryotic (left) and (right) *S. solfataricus* primases. **b** Primer initiation complexes for eukaryotic PriSL (top), archaeal PriSLX (middle) and Chimera (bottom)

**Poly(dT)-templated primase assay**. Ten-microlitre reactions contained 50 mM MES-NaOH pH 6.5, 10 mM $MnCl_2$, 0.1 mg ml$^{-1}$ BSA, 10 μM $ZnCl_2$, 10 μM ATP, 1 μCi [α-$^{32}$P]ATP, 100 ng poly(dT) (Amersham biosciences), 150 μM ApA (TriLink BioTechnologies, presented in ApA extension assay), 160 nM ribo(A8) (IDT, presented in 8mer primer extension assay) and the indicated amount of primases. The reactions were incubated at 75 °C for 20 min before being quenched by the addition of an equal volume of loading buffer (98% formamide, 10 mM EDTA). The mixtures were boiled for 5 min and analyzed by electrophoresis through a 20% polyacrylamide, 8 M Urea, 1 × TBE gel and visualized by phosphorimagery on a Typhoon scanner (GE Healthcare).

**End-labeled primer extension assay**. ApA or ribo(A8) was end labeled with [γ-$^{32}$P]ATP using T4 PNK (New England Biolabs) and labeled ribo(A8) primer was annealed to poly(dT) prior to the reaction. The reaction condition is essentially the same as the primase assay except containing 80 μM ATP and $^{32}$P 5′-end-labeled ApA (50 nM) or ribo(A8) primed poly(dT) (33 nM). ApA and ribo(A8) primer extension products were analyzed with 23% and 20% polyacrylamide gel, respectively.

**M13 DNA replication assay**. Reactions were performed in a volume of 10 microlitres containing 50 mM Tris pH 6.0, 10 mM DTT, 0.1 mg ml$^{-1}$ BSA, 10 mM $MgCl_2$, 200 μM each rCTP, rUTP, rGTP, 5 mM ATP, 60 μM of each dATP, dTTP, dGTP, 20 μM dCTP, 2 μCi [α-$^{32}$P]dCTP, 5 ng M13mp18 ssDNA (New England Biolabs), 24 nM MCM, 140 nM Cdc45-GINS, 100 nM PCNA, 50 nM RFC, 50 nM PolB1-HE, 400 nM SSB and 50 nM primase (unless otherwise indicated). The reactions were incubated at 75 °C for 20 min before being quenched by the addition of an equal volume of stop buffer (40 mM EDTA, 1% SDS) and 4 μl alkaline gel-loading buffer (300 mM NaOH, 6 mM EDTA, 18% (w/v) Ficoll, 0.15% (w/v)

bromocresol green, 0.25% (w/v) xylene cyanol). The products were analyzed by electrophoresis through a 1% alkaline agarose gel and visualized using by phosphorimaging on a Typhoon scanner.

**Fluorescence anisotropy**. The nucleotide-binding affinity of PriX was determined by measuring fluorescence polarization of fluorescein-labeled ATP (Jena Bioscience) in the presence of increasing amounts of PriX. The labeled ATP was used at a constant concentration of 20 nM and mixed with a dilution series of PriX ranging from 0 to 20–30 μM concentration, in 25 mM Hepes 7.2, 150 mM NaCl, 2 mM MnCl$_2$, 1 mM TCEP buffer. A total of 90 μL of each sample was transferred into 96-well NBS plates (Corning) and the polarization measured with a PHER-Astar FS plate reader (BMG Labtech) using an excitation wavelength of 485 nm and an emission wavelength of 520 nm at 25 °C.

**Analytical BMOE cross-linking**. A concentration of 2.2 mg of BMOE (Thermo Fisher Scientific) were dissolved in 0.5 mL DMSO to obtain a 20 mM cross-linker stock. A 200-fold dilution (0.1 mM) with reaction buffer (25 mM Hepes pH 7.2, 150 mM NaCl, 5 mM MnCl$_2$) was prepared and 2 μL of it added to 17 μL of 5 μM primase. Depending on the experiment, 10 μM of a DNA/RNA hybrid with or without a 5′-triphosphate was added to the primase solution (5′-ppp-(GGCUC GG)$_{RNA}$-3′ and 5′-(AACACCGAGCCAACAT)$_{DNA}$-3′). The cross-linking reaction was incubated at 25 °C for 1 h. To quench the reaction, 1 μL of 1 M DTT was added to the reaction mixture and incubated for an additional 15 min. Finally, 10 μL of SDS gel-loading buffer was added and 15 μL of the mixture loaded onto a SDS-PAGE gel for analysis.

**Preparative BMOE cross-linking**. The individual and double PriS E34C and PriX K110C mutants of Chimera were prepared as described above and adjusted to a protein concentration of 20 μM. BMOE to a final concentration of 40 μM was then added to the protein sample and incubated for 30 min at 25 °C before the absolute BMOE concentration in the reaction was increased to 80 μM and incubated for further 30 min. Subsequently the reaction was stopped by adding a final concentration of 5 mM DTT to the mixture followed by a 10 min incubation at 25 °C. If necessary, the protein was concentrated with a spin concentrator to a volume of 1–2 mL and subjected to a Superdex S200 16/600 size-exclusion chromatography column (GE Healthcare). The purified cross-linked protein was then used for gel-based activity assay as described above.

**Data availability**. Coordinates and structure factors have been deposited in the Protein Data Bank under accession codes 5OF3 (PriSLX) and 5OFN (Chimera). The data that support the findings of this study are available from the corresponding author upon request.

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

## Acknowledgements

We thank Joseph Maman for helpful discussions. This work was supported by a Wellcome Trust investigator award to L.P. (104641/Z/14/Z), a PhD fellowship of the Boehringer-Ingelheim Fonds and awards from the Janggen-Pöhn-Stiftung and the Swiss

National Science Foundation to S.H. SDB's laboratory is funded by the College of Arts and Sciences, Indiana University.

## Author contributions

S.H. solved the crystal structures of PriSLX and Chimera, prepared recombinant PriSLX proteins and Chimera (wild-type and mutants), and performed the primase assays on mixed-sequence DNA template, including those with cross-linked Chimera; J.Y. purified the *S. solfataricus* replication proteins, performed all primase assays on poly(dT) DNA template and the DNA replication assays. S.H. and S.D.B. and L.P. conceived the project; S.H., J.Y., M.L.K., S.D.B., and L.P. designed the experiments; L.P. wrote the paper, with input from all authors.

## Additional information

**Competing interests:** The authors declare no competing financial interests.

