## [Peer Review File · Nature Communications]

Reviewers' comments:

Reviewer #1 (Remarks to the Author):

Two major findings of current work are the discovery of initiation site structure with bound AMPCPP and showing that Fe-S cluster in PriL plays only structural role and is dispensable when PriX is fused to truncated PriL via a flexible linker.

Because initiation site is known for human primase, detailed comparison of initiation site residues with bound nucleotides in two structures is highly advised. Is there any 3-d conservation of arginines and tyrosine interacting with triphosphates?

The Figure 9 and conclusions will be easier to understand if one more figure will be added with superimposition of eukaryotic PriL-CTD and PriX showing the location of Fe-S cluster and AMPCPP.

Other points

Remove Fig 2a and show the positions of AMPCPP in Fig 1a.

Fig 2b needs to be revised to show all interactions of AMPCPP with PriX. Better with distances and/or stereo.

Fig 3a is dispensable.

Reviewer #2 (Remarks to the Author):

The manuscript written by Holzer et al describes the crystal structure of the archaeal primase complex, consisting of PriS, PriL and PriX, originated from the hyperthermophilic archaeon, *Sulfolobus solfataricus*. They identified one each nucleotide binding site, which are important for initiation and elongation of the primer synthesis, in PriS and PriX, respectively. Another interesting point of the study is that the C-terminal region containing the S-Fe cluster in the PriL subunit can be substituted by PriX. These data provide the knowledge to understand the concrete function of PriX, the third subunit of the archaeal primase, and why it is essential for viability.

I think this work is well organized and experimental data are mostly well presented. Their results are generally interesting, especially for the readers in the field of DNA replication and are valuable for publication. I have some comments, which I hope will be helpful to improve the manuscript before publication.

1. Identification of the nucleotide-binding site in PriX is a very important topic of this study, because this nucleotide binding determines the initiation of the primer synthesis reaction. The authors closed up the structure of the nucleotide binding in Fig. 2, and it is understandable that D70, R72, and R74 are important to interact with ATP. However, I think the readers want to know how to determine the preferable nucleotide for the initiation of primer synthesis and why ribonucleotides, rather than deoxyribonucleotides, are selected. Furthermore, the authors focused on the AMPCPP binding site in PriS in Suppl Fig. 2 and discussed the reason why ribonucleotide is selectively incorporated for the elongation reaction. They predicted that the interaction of the 2'-OH of the nucleotide with L245 and R246 by hydrogen bonding is a reason why the primase prefers to use ribonucleotide. I think the Suppl Fig 2 should be moved to the main part and should be drawn to show these interactions more clearly. Fig. 3A is redundant and is not needed to be shown. Typo; "The nucleotide is bound is" in page 5 line 5 from the bottom.

2. Figure 7 showed their structure-guided site-specific cross-linking experiment. I think the predicted position of T11, even though it is located in the disordered region, should be indicated in Fig. 7A to help readers to understand the results shown in Fig. 7B. In addition, the other protein bands produced on the gel image by the cross-linking reaction should be assigned as much as possible.

3. The beginning part of the legend to Fig. 8 does not match to the figure. The authors described that the cross-linked primase complex was separated by gel filtration and the fused protein was isolated. I think the gel filtration profiles and the PAGE image of the isolated complex should be included with these primer synthesis assays. In addition, observations of the different extension products between Fig 8A, 8B, and 8C should be explained in more detail. For example, why was the 20 mer primer not extended in 8A? Also, why was the 18 mer RNA primer extended in the presence of NTP in 8B, but not in the presence of dNTP in 8C? Why did the cross-linked primase extend 18 mer primer more efficiently than other primases in 8B, and also why was this more efficient extension of the cross-linked primase not observed in the presence of dNTP in 8C?

4. There are many archaeal organisms lacking the PriX subunit. I think some description of the distribution of the primase subunits in Archaea and sequence comparison of the PriL subunits among the archaea possessing and not possessing PriX will be helpful for the readers to understand the importance of this work.

More minor comments

1. Explanation of the DNA/RNA size makers on the gels in Fig. 3, Fig. 4, Fig. 5, Fig. 6, and Fig. 8 should be added. For example, 30 mer template is bigger than the size marker of 30 mer in Fig. 8A, and also 18 mer primer is the same as the size marker of 20 mer in Fig. 8B, and C.

1. The sizes of the marker proteins indicated in the left side of the gel images in Fig. 7B, Suppl Fig. 1B, and Suppl Fig. 6 should not be written in kDa. The molecular weights should be expressed by (x 103), but not [kDa], because molecular weight is dimensionless.

Reviewer #3 (Remarks to the Author):

In their manuscript titled “Primer synthesis without an Fe-S cluster by a eukaryotic-like archaeal primase”, the authors report structural and biochemical characterization of the heterotrimeric PriSLX primase from the archaeon *Sulfolobus solfataricus*. The structure of PriSLX determined to 2.9 Å represents neither the initiation nor the elongation complex, and probably captures an intermediate conformation. Nonetheless, the reported structure aids the identification of a NTP binding site on PriX that is similar to the putative NTP site on PriL. Enzymatic activity assays to test the role of the NTP binding site on primer synthesis indicate that the NTP site is essential for initiation of primer synthesis *in vitro*. In addition, a chimeric construct of PriSLX, lacking the Fe-S cluster of PriL is as active as the wild type heterotrimer in primer synthesis. The work reported is technically sound and original. Overall, the manuscript is well-written and suitable for publication in *Nature Communications* but requires modifications.

Comments:

1. On Pg. 3 where PrimPol is referenced, the authors should cite the following publication elucidating the only available structure of PrimPol bound to DNA template-primer duplex: Rechkoblit O., et al. Structure and mechanism of human PrimPol, a DNA polymerase with primase activity. *Sci. Adv.* 2016 Oct 21;2(10):e1601317

2. On Pg. 4, last line of introduction, the authors should clarify that the Fe-S cluster is dispensable for primer synthesis in *S. solfataricus*.

3. The manuscript is missing any attempts to describe the conformation of PriL-CTD and the Fe-S cluster. It may be helpful to include a description of this domain in the Results section and in Figure 1. Additionally, in Figure 1, it may help to clearly label the N and C-terminus of PriS, PriL, and PriX and highlight the active site residues.

4. On Pg. 10, the authors state, "...the archaeal/eukaryotic primases possess two independent NTP binding sites, the elongation and initiation sites, that must become transiently juxtaposed...". It is hard to see how PriS and PriX can juxtapose to form the initial dinucleotide. Is it feasible to juxtapose the PriS and PriX NTP binding sites without steric interference?

5. In the first paragraph on Pg. 10, the authors note "...highlight a distinctive mechanism of initial dinucleotide synthesis for primase, different for instance from the initiation mechanism employed by transcription RNA polymerases, where the active site is buried deep within the multi-subunit enzyme and can accommodate two NTPs during transcription initiation." A mechanism for binding 2 NTPs in a single active site has been suggested for PrimPol (Rechkoblit O., et al). The authors should reference this in the discussion.

6. On Pg. 11, the authors may want to note the absence of PriX homologs in eukaryotes

7. On Pg. 11, the authors draw parallels between the archaeal and eukaryotic primases, suggesting, "...it is unlikely that the Fe-S cluster would have a redox function in priming reactions of the eukaryotic enzyme." This is not entirely accurate, since the eukaryotic enzyme works in combination with Pol α and could possibly use a different mechanism. In which case, the authors may want to tone down the equivalences between the archaeal and eukaryotic primases in the manuscript (perhaps even in the title).

Reviewer #1 (Remarks to the Author):

Two major findings of current work are the discovery of initiation site structure with bound AMPCPP and showing that Fe-S cluster in PriL plays only structural role and is dispensable when PriX is fused to truncated PriL via a flexible linker.

Because initiation site is known for human primase, detailed comparison of initiation site residues with bound nucleotides in two structures is highly advised. Is there any 3-d conservation of arginines and tyrosine interacting with triphosphates?

As discussed in our manuscript (second paragraph of page 6), comparison of nucleotide-bound PriX with human and yeast PriL-CTDs identified arginine residues 72 and 74 as potential functional analogues of R355 and R306 in yeast and human PriL-CTD respectively, which are known to be critical for primer synthesis (refs 12, 19 and 22 of our paper). No obvious structural conservation of tyrosine residues was observed, although PriX Y103 contacts the triphosphate of the AMPCPP (Figure 2B) and thus might fulfill a similar role to Y345 in the structure of human PriL-CTD bound to RNA/DNA (ref. 25).

The Figure 9 and conclusions will be easier to understand if one more figure will be added with superimposition of eukaryotic PriL-CTD and PriX showing the location of Fe-S cluster and AMPCPP.

As requested by the reviewer, we have added a panel to Supplementary figure 4 of the revised manuscript (now panel B), showing a superposition of PriX with yeast and human PriL-CTD, highlighting the relative positions of AMPCPP and the Fe-S cluster. An in-depth discussion of the structural similarity between PriX and the eukaryotic PriL-CTDs can be found in Liu et al, *Nat Comm*, 2015 (ref. 26 of our manuscript).

Other points

Remove Fig 2a and show the positions of AMPCPP in Fig 1a.

The purpose of Figure 1 is to show the overall architecture of the PriSLX heterotrimer, whereas Figure 2A illustrates the position of the two nucleotide analogue AMPCPP molecules bound to PriS and to PriX, and therefore marks the relative location of the elongation and initiation sites in PriSLX. This is a central finding of the paper and we believe that it is preferable to illustrate it in this way.

Fig 2b needs to be revised to show all interactions of AMPCPP with PriX. Better with distances and/or stereo.

In accordance with the reviewer's request, we have re-drawn Fig. 2B, so that all polar contacts between PriX and AMPCPP are now explicitly drawn, with distances. The figure shows all PriX residues within 5Å of the AMPCPP molecule, except Q69 and E142, which do not contact the AMPCPP. We would prefer not to show stereo drawings as most readers don't really benefit from them.

Fig 3a is dispensable.

We have removed Fig. 3A, as requested by the reviewer.

Reviewer #2 (Remarks to the Author):

I think this work is well organized and experimental data are mostly well presented. Their results are generally interesting, especially for the readers in the field of DNA replication and are valuable for publication. I have some comments, which I hope will be helpful to improve the manuscript before publication.

1. Identification of the nucleotide-binding site in PriX is a very important topic of this study, because this nucleotide binding determines the initiation of the primer synthesis reaction. The authors closed up the structure of the nucleotide binding in Fig. 2, and it is

understandable that D70, R72, and R74 are important to interact with ATP. However, I think the readers want to know how to determine the preferable nucleotide for the initiation of primer synthesis and why ribonucleotides, rather than deoxyribonucleotides, are selected. As we describe in the text and in Figure 2, our structure shows that the ribose of AMPCPP, including its 2'-OH, is exposed to solvent and not involved in interactions with PriX. It is possible that NTP selection happens in the 'initiation complex' of primase, during the dinucleotide synthesis step. Furthermore, assuming higher cellular concentrations of NTP than dNTP in archaea as observed in bacteria and eukaryotes, a ribonucleotide will be preferentially bound in the initiation (as well as the elongation) site of primase. We have added a brief comment in the text concerning this point.

Furthermore, the authors focused on the AMPCPP binding site in PriS in Suppl Fig. 2 and discussed the reason why ribonucleotide is selectively incorporated for the elongation reaction. They predicted that the interaction of the 2'-OH of the nucleotide with L245 and R246 by hydrogen bonding is a reason why the primase prefers to use ribonucleotide. I think the Suppl Fig 2 should be moved to the main part and should be drawn to show these interactions more clearly. Fig. 3A is redundant and is not needed to be shown. Typo; "The nucleotide is bound is" in page 5 line 5 from the bottom.

We would prefer to keep Supplementary figure 2 in the supplementary information, as the focus of the paper is really on the identification of the initiation site in PriX, and our manuscript already contains 9 figure items. A similar set of contacts involving the 2'-hydroxyl of the ribose was observed in our structure of human PriSL bound to UTP (Kilkenny et al, PNAS, 2013; ref. 12 of our manuscript), and we now comment briefly on this point in the revised manuscript.

As requested by the reviewer, we have removed Fig. 3A, and corrected the typo.

2. Figure 7 showed their structure-guided site-specific cross-linking experiment. I think the predicted position of T11, even though it is located in the disordered region, should be indicated in Fig. 7A to help readers to understand the results shown in Fig. 7B. In addition, the other protein bands produced on the gel image by the cross-linking reaction should be assigned as much as possible.

In accordance with the reviewer's request, we have prepared a new version of Figure 7A, where we show the position of PriS T11. We have also annotated further the gel image of Figure 7B, by indicating the extent and position of the higher-molecular weight species resulting from heterogeneous crosslinking in the control reactions with the T11C PriS mutant.

3. The beginning part of the legend to Fig. 8 does not match to the figure. The authors described that the cross-linked primase complex was separated by gel filtration and the fused protein was isolated. I think the gel filtration profiles and the PAGE image of the isolated complex should be included with these primer synthesis assays.

We have now added the requested information, in Supplementary figure 7 of the revised manuscript.

In addition, observations of the different extension products between Fig 8A, 8B, and 8C should be explained in more detail. For example, why was the 20 mer primer not extended in 8A?

Figure 8A shows the result of an initiation assay, in which we tested the ability of the crosslinked Chimera to synthesise an RNA primer *de novo*, and therefore no primer is added to the reaction (see input lane). The band of about 20 nucleotides visible in the gel represent the RNA product synthesized by WT and single-mutant controls, but not by the crosslinked protein. We have annotated Figure 8 to clarify the nature of the reaction products in the three panels.

Also, why was the 18 mer RNA primer extended in the presence of NTP in 8B, but not in the presence of dNTP in 8C? Why did the cross-linked primase extend 18 mer primer more efficiently than other primases in 8B, and also why was this more efficient extension of the cross-linked primase not observed in the presence of dNTP in 8C?

Figure 8B, C shows that the crosslinked primase is able to extend the RNA primer, with both NTP and dNTP. As the reviewer points out, panel B shows that there are some differences in the product profile of the crosslinked Chimera, relative to the other control samples. We attribute these differences to the fact that, in the presence of NTP, the control reactions contain a composite of initiation and extension products, whereas the crosslinked protein can only extend the existing primer, but cannot initiate a new primer.

Panel C shows only primer extension products for all samples, as primase cannot initiate, or very poorly, with dNTP. In all reactions, a predominant extension of a few nucleotides in length is observed. At the moment, we are unclear as to the reason of the small difference in extension between the crosslinked Chimera and the single-mutant protein controls, although it is interesting to observe that the extension products of the crosslinked Chimera seems to be the same as those of the WT protein. We note that the extent of the extension reaction could also be affected by the relatively short template DNA used (38 nucleotides).

4. There are many archaeal organisms lacking the PriX subunit. I think some description of the distribution of the primase subunits in Archaea and sequence comparison of the PriL subunits among the archaea possessing and not possessing PriX will be helpful for the readers to understand the importance of this work.

Liu et al, *Nat Comm*, 2015 (ref. 26) have already provided a detailed analysis of the evolutionary distribution of PriX in archaea (Figure 3 and 4 of their paper), and we believe that it would be superfluous to add to their exhaustive commentary on this.

More minor comments

1. Explanation of the DNA/RNA size makers on the gels in Fig. 3, Fig. 4, Fig. 5, Fig. 6, and Fig. 8 should be added. For example, 30 mer template is bigger than the size marker of 30 mer in Fig. 8A, and also 18 mer primer is the same as the size marker of 20 mer in Fig. 8B, and C.

The template DNA 30mer used in the experiments of Figure 8A runs on average as the 30mer band of the markers across all samples, with small variations in mobility that are due to differences in sequence and slight differences in the polyacrylamide gel.

The primer 18mer used in Figure 8B, C is made of RNA, and therefore has a slightly different electrophoretic mobility than the DNA 20mer used as size marker, as well as a different nucleotide sequence. Similar explanations apply to gels in Figs 3-6 where nucleic acid markers are used.

1. The sizes of the marker proteins indicated in the left side of the gel images in Fig. 7B, Suppl Fig. 1B, and Suppl Fig. 6 should not be written in kDa. The molecular weights should be expressed by (x 103), but not [kDa], because molecular weight is dimensionless.

We have modified the relevant figures as requested by the reviewer.

Reviewer #3 (Remarks to the Author):

The work reported is technically sound and original. Overall, the manuscript is well-written and suitable for publication in Nature Communications but requires modifications.

Comments:

1. On Pg. 3 where PrimPol is referenced, the authors should cite the following publication elucidating the only available structure of PrimPol bound to DNA template-primer duplex:

Rechkoblit O., et al. *Structure and mechanism of human PrimPol, a DNA polymerase with primase activity*. *Sci. Adv.* 2016 Oct 21;2(10):e1601317

We have added the reference, as requested by the reviewer.

2. On Pg. 4, last line of introduction, the authors should clarify that the Fe-S cluster is dispensable for primer synthesis in *S. solfataricus*.

We would like to point out that, in the sentence referred to by the reviewer, we use the words 'its Fe-S cluster' to make clear that we refer to the archaeal primase of *S. solfataricus*.

3. The manuscript is missing any attempts to describe the conformation of PriL-CTD and the Fe-S cluster. It may be helpful to include a description of this domain in the Results section and in Figure 1. Additionally, in Figure 1, it may help to clearly label the N and C-terminus of PriS, PriL, and PriX and highlight the active site residues.

The PriL-CTD was poorly ordered in our electron density map and only portions of it, which did not include the Fe-S cluster, could be built into the crystal structure. The weak definition of the PriL-CTD in our map is due to its flexible connection to the rest of PriL; a similar degree of flexibility and crystallographic disorder of the PriL-CTD was observed in the crystal structure of human primase (Baranovskiy et al, *JBC*, 2015). Consequently, we did not include a detailed structural description of our archaeal PriL-CTD in the manuscript.

To help the reader identifying the PriL-CTD in our structure, we have now annotated its position in the structural drawing of Figure 1. The position of the Fe-S cluster in the eukaryotic PriL-CTD is also indicated in Supplementary figure 4B of the revised manuscript. To further assist the reader, we refer in our Introduction to several papers (Sauguet et al, *PLoSOne*, 2010; Vaithiyalingam et al, *PNAS*, 2010; Agarkar et al, *Cell Cycle*, 2011) where the structure of the eukaryotic PriL-CTD and its Fe-S cluster is described in detail.

As requested by the reviewer, we have labelled the N- and C-termini of PriS, PriL and PriX and marked the active site of PriS with a star.

4. On Pg. 10, the authors state, "...the archaeal/eukaryotic primases possess two independent NTP binding sites, the elongation and initiation sites, that must become transiently juxtaposed...". It is hard to see how PriS and PriX can juxtapose to form the initial dinucleotide. Is it feasible to juxtapose the PriS and PriX NTP binding sites without steric interference?

As we explain in the Discussion, we believe that our PriSLX structure captures a primase state that is competent for primer elongation but not initiation. It is reasonable to assume therefore that a large structural rearrangement must take place, to reposition the initiation site in PriX alongside the active site of PriS, for synthesis of the initial di-nucleotide. As PriX is only flexibly tethered to PriL-CTD and doesn't engage in extensive interactions with PriS, we see no steric impediment to such rearrangement, the exact nature of which remains unknown and has been speculated upon extensively in the literature (see for instance refs 12, 13, 23 and 25).

5. In the first paragraph on Pg. 10, the authors note "...highlight a distinctive mechanism of initial dinucleotide synthesis for primase, different for instance from the initiation mechanism employed by transcription RNA polymerases, where the active site is buried deep within the multi-subunit enzyme and can accommodate two NTPs during transcription initiation." A mechanism for binding 2 NTPs in a single active site has been suggested for PrimPol (Rechkoblit O., et al). The authors should reference this in the discussion.

As requested by the reviewer, we added the reference to our Discussion.

6. On Pg. 11, the authors may want to note the absence of PriX homologs in eukaryotes

In the Introduction, we make clear that PriX is found only in Archaea. We have also modified the second sentence in the penultimate paragraph of the Discussion, to emphasise that PriX has functionally replaced PriL-CTD in the archaeal primase of *S. solfataricus*.

7. On Pg. 11, the authors draw parallels between the archaeal and eukaryotic primases, suggesting, "...it is unlikely that the Fe-S cluster would have a redox function in priming reactions of the eukaryotic enzyme." This is not entirely accurate, since the eukaryotic enzyme works in combination with Pol α and could possibly use a different mechanism. In which case, the authors may want to tone down the equivalences between the archaeal and eukaryotic primases in the manuscript (perhaps even in the title).

We agree with the reviewer that it remains formally possible that the eukaryotic primases operate by a different mechanism of primer synthesis, which requires the presence of a redox-active Fe-S cluster, and we have modified the concluding remarks of our Discussion as well as the title to highlight that our evidence concerns the archaeal enzyme.

Given the clear evidence presented in our paper though, one would have to assume that archaeal and eukaryotic primases - closely related in sequence, structure and biochemical function - had diverged substantially when it comes to the role of their Fe-S cluster, so that it would have become dispensable in archaeal primases whereas in eukaryotic primase it would have maintained an essential redox role. Although strictly speaking this remains a possibility, we believe that it is a remote one.